# Blood Glucose, Lactate and Platelet Count in Infants with Spontaneous Intestinal Perforation versus Necrotizing Enterocolitis—A Pilot Study

**DOI:** 10.3390/children10061028

**Published:** 2023-06-08

**Authors:** Jacky Herzlich, Dror Mandel, Ronella Marom, Rafael Mendelsohn, Audelia Eshel Fuhrer, Laurence Mangel

**Affiliations:** 1Department of Neonatology, Dana Dwek Children’s Hospital, Tel Aviv Sourasky Medical Center, Tel Aviv 6423906, Israel; 2Sackler Faculty of Medicine, Tel Aviv University, Tel Aviv 6997801, Israel; 3Department of Pediatric Surgery, Dana Dwek Children’s Hospital, Tel Aviv Sourasky Medical Center, Tel Aviv 6423906, Israel

**Keywords:** spontaneous intestinal perforation, necrotizing enterocolitis, preterm infant, blood lactate, blood glucose

## Abstract

The incidence of spontaneous intestinal perforation (SIP) increases up to 10% with decreasing gestational age (GA). We aimed to explore early biomarkers for predicting SIP in preterm infants. In this case–control study, neonates born at ≤34 weeks GA diagnosed with SIP were compared with GA and/or birth-weight-matched neonates diagnosed with necrotizing enterocolitis (NEC). Laboratory markers assessed prior and adjacent to the day of SIP or NEC diagnosis were evaluated. The cohort included 16 SIP and 16 matched NEC infants. Hyperlactatemia was less frequent in SIP than in NEC infants (12% vs. 50%, *p* = 0.02). The platelets count was lower in SIP than in NEC infants (*p* < 0.001). Glucose levels strongly correlated with lactate levels (*p* = 0.01) only in the NEC group. The odds of being diagnosed with SIP decreased as lactate levels increased (OR = 0.607, 95% CI: 0.377–0.978, *p* = 0.04). Our results suggest that a combination of laboratory markers, namely glucose and lactate, could help differentiate SIP from NEC at early stages so that, in the presence of an elevated blood glucose, an increase in blood lactate was associated with a decrease in the odds of being diagnosed with SIP.

## 1. Introduction

Spontaneous intestinal perforation (SIP) of the preterm infant is defined as an isolated intestinal perforation typically found at the terminal ileum [1,2]. SIP occurs primarily in preterm infants with very low birth weight (VLBW, birth weight < 1500 g) and extremely low birth weight (ELBW, birth weight < 1000 g) with a reported incidence of 2–3% and up to 10%, respectively [1,3,4,5]. SIP occurs more frequently in male infants (65 to 71% of affected infants) [1,4]. Although SIP is a separate clinical entity different from the most severe gastrointestinal complication of prematurity, necrotizing enterocolitis (NEC), both entities might have overlapping presentations with similar physical and X-ray findings [1,4,5,6,7]. The etiology of SIP remains poorly understood. While several antenatal and postnatal risk factors have been suggested, prematurity and very low birth weight are the main established risk factors [8,9]. Several studies have discussed the possibility of misdiagnosis of SIP as NEC [3,10]. There is growing literature on laboratory markers in the early diagnosis of NEC and in distinguishing NEC from other non-NEC intestinal pathologies [11,12,13].

Yet, there is a paucity of data on laboratory markers for predicting SIP. A large prospective multicenter study found only a moderate ability to distinguish the isolated intestinal perforation from NEC entities pre-operatively [14]. Moreover, the management of SIP may differ from NEC, while some surgeons may be more likely to treat a suspected SIP with a peritoneal drain over laparotomy [15]. Early distinction between these entities may provide opportunities for a more tailored surgical approach. We therefore designed the current study in order to assess the predictive abilities of laboratory parameters in early diagnosis of SIP in very low to extremely low birth weight preterm infants when compared to NEC. We hypothesized that laboratory markers may differ between SIP and NEC at early stages of the disease.

## 2. Materials and Methods

### 2.1. Study Design and Participants

This is a retrospective case–control study of neonates, born at ≤34 weeks gestational age (GA) diagnosed with SIP or NEC, between 1 January 2010 and 31 March 2021, at the Lis Maternity Hospital of the Tel Aviv Medical Center and admitted to the Neonatal Intensive Care Unit of this tertiary referral hospital. SIP infants were matched, in a 1:1 ratio, with medical or surgical NEC infants, by GA (GA ± 2 weeks) and/or birth weight (BW ± 15%). We excluded infants at GA > 34 weeks and infants with an intestinal pathology other than NEC or SIP. The study was approved by the local institutional review board of the Tel Aviv Medical Center (0028-21-TLV). Informed consent was waived due to the retrospective nature of the study.

### 2.2. Data Collection and Assessment

A single investigator (J.H.) reviewed the extracted hospital medical records using the “intestinal perforation” keyword, to confirm infants’ prematurity and diagnosis of either SIP or NEC. NEC cases were defined by modified Bell’s criteria of stage 2 or greater with radiographic evidence of pneumatosis intestinalis [16]. Physical examination and X-ray imaging confirmed the diagnosis of SIP with radiographic evidence of pneumoperitoneum. Maternal collected data included age, prenatal medication and conditions. Demographic and clinical data of preterm infants included GA, gender, BW, mode of delivery, medications, Apgar scores (1 and 5 min), laboratory parameters and treatment of condition. Per hospital guidelines, all infants in this study were fed according to standardized Neonatal Intensive Care Unit (NICU) feeding protocol with an exclusive human milk-based diet when possible and complementary infant formula when necessary. Blood analyses were performed following our NICU standard of care protocol (at admission, within the first week of life, every 7 days when the baby is stable and at discharge). Following case matching, we recorded laboratory parameters (glucose, lactate, platelet, white blood cell (WBC) and nucleated red blood cells (NRBC)), obtained during routine care, that were performed prior and adjacent (12 to 24 h) to the diagnosis of SIP or NEC. There is a lack of consensus as to what the threshold for neonatal hyperglycemia in VLBW or ELBW infants is [17,18]. Therefore, we recorded both mild to moderate hyperglycemia (blood glucose > 120 mg/dL) and severe hyperglycemia (≥180 mg/dL) as events of hyperglycemia [17]. Appropriate, small and large for gestational age newborns (AGA, SGA, LGA) were defined according to the Israeli-specific intrauterine growth charts [19].

### 2.3. Statistical Analysis

IBM SPSS Statistics for Windows, version 25, was used for statistical data analyses. Categorical variables were reported as frequencies and percentages. Normality of continuous variables was assessed by Shapiro–Wilk tests. Data were expressed as mean ± standard deviation (SD) and range for normally distributed variables and median and interquartile range (IQR) for skewed distributions. Chi square tests or Fisher’s exact tests were applied to compare categorical variables and the Mann–Whitney U test to compare continuous variables, between the SIP and the NEC groups. Pearson’s correlation coefficient was used to evaluate the relationship between the assessed variables. Adjusted odds ratios (ORs) were calculated with 95% confidence intervals (CIs) for the association of laboratory parameters (i.e., lactate, glucose, platelets) with the diagnosis of SIP, in logistic regression. All *p* values were 2-sided and *p* < 0.05 was considered statistically significant.

## 3. Results

As depicted in Figure 1, 282 eligible infants were retrieved; 16 (5.7%) were preterm infants diagnosed with SIP, 163 (57.8%) with NEC and the remaining 103 (36.5%) were neither SIP, NEC nor preterm infants. All 16 cases of SIP were included in the analysis along with 16 cases of NEC matched by GA and/or BW.

Descriptive characteristics of mothers and infants in the SIP and NEC groups are presented in Table 1. Per design, preterm infants in both groups had similar GA and BW. The NEC group contained 4 surgical NEC and 12 medical NEC. The SIP and the NEC groups did not differ in terms of maternal age, mode of delivery, male–female ratio, percentage of LGA, AGA, or SGA infants, Apgar scores (1 and 5 min), percentage of pre-eclampsia, placental abruption, the use of prenatal steroids, magnesium, maternal antibiotics, the use of inotropic agent, ibuprofen, fresh frozen plasma and mortality. Although the rate of chorioamnionitis was much higher in the SIP group, this difference was not statistically significant. In addition, both groups had similar rates of exclusive human milk feeding. All SIP infants underwent an intervention, either stoma (*n* = 9), primary closure (*n* = 3) or drainage (*n* = 4) as opposed to NEC infants (3 stoma and 1 explorative laparotomy) (*p* < 0.001). The majority of NEC infants (75%) were managed expectantly. Five neonates died in the SIP group, all at GA ≤ 28 week, two of them shortly after diagnosis and surgical treatment (drainage or stoma), the other three due to complications of prematurity. In the NEC group, three out of the four deaths occurred following surgical treatment. Platelets count (median [IQR], K/µL) was lower in the SIP group (130.5 [97.2–184.8]) than in the NEC group (347.5 [186.8–510.2]), (*p* < 0.001). Hence, the incidence of thrombocytopenia (platelet count < 150 K/µL) was significantly higher in the SIP group (56.3%) than in the NEC group (12.5%), (*p* = 0.009). Median nucleated red blood cells (NRBC) count was higher in SIP infants than in NEC, but did not reach significance. Hyperglycemia (Glucose level > 120 mg/dL) was present in most of the SIP (88%) and NEC (81%) infants, whereas hyperlactatemia (Lactate level > 4 mmol/L) was less frequent in the SIP group than in the NEC group (12.5% versus 50%, respectively, *p* = 0.022). However, recorded blood lactate levels were not statistically different between the groups (*p* = 0.109). SIP was diagnosed within the first days of life as opposed to NEC (median [IQR], 5 [3.2–7.8] vs. 19 [11–29.5], *p* < 0.001).

A Pearson correlation coefficient was computed to assess the relationship between glucose and lactate levels in blood. There was a strong positive correlation between these two variables in the NEC group (*r* = 0.612; 95% CI: 0.168–0.850, *p* = 0.012) and an absence of correlation in the SIP group (*r* = −0.265; 95% CI: −0.672–0.265, *p* = 0.321) where only glucose levels increased (Figure 2). A logistic regression was performed to ascertain the effects of laboratory parameters (platelet, lactate and glucose) on the likelihood that infants will be diagnosed with SIP. The logistic regression model was statistically significant, χ^2^(3) = 21.175, *p* < 0.001. The model explained 64.5% of the variance in SIP diagnosis and correctly classified 78.1% of cases. Every one-unit increase in lactate was associated with a 39% decrease in the odds of being diagnosed with SIP, and every 1 K/µL increase in platelet was associated with a 2% decrease in the odds of being diagnosed with SIP (Table 2).

## 4. Discussion

In this study, we showed that in preterm infants diagnosed with SIP, while glucose levels increased, lactate levels remained within the normal range, at early stage prior to clinical manifestation leading to diagnosis of perforation. Conversely, in NEC, an elevation in glucose levels was usually coupled with an elevation in lactate levels prior to clinical manifestation leading to NEC diagnosis (medical or surgical). NEC can be lethal in VLBW and ELBW infants. There are continuing improvements in neonatal care and surgical outcomes for abdominal pathologies. Early detection is critical to that process. NEC and SIP might have overlapping clinical signs at their initial presentation [6,10]. Srinivasjois et al. showed that plasma glucose and lactate levels were significantly higher when compared with baseline levels, at all-time points during NEC [11]. Additionally, plasma lactate and glucose levels may predict progression of definite NEC to surgery or death in preterm neonates [11]. Elevated plasma lactate has been strongly associated with the worst prognosis in NEC [20]. In our study, the NEC group showed blood lactate and glucose levels consistent with Srinivasjois et al.’s findings [11]. Yet, the ability of laboratory markers as predictors of SIP has not been fully investigated. Recently, Barseghyan et al. showed that a transient increase in serum alkaline phosphatase level was independently associated with SIP when compared to NEC [21]. Shah et al. have demonstrated that blood levels of inter-alpha inhibitor proteins may assist in early detection of NEC and distinguish NEC from SIP and other nonspecific abdominal disorders of preterm infants [22].

Our study showed that a combination of laboratory markers, namely glucose and lactate, could help differentiate SIP from NEC, at early stages, so in the presence of an elevated blood glucose, an increase in blood lactate was associated with a decrease in the odds of being diagnosed with SIP. The sudden increase in blood glucose in SIP infants occurred within the 24 h prior to perforation. Elevated blood lactate levels are usually a marker for tissue alteration [20]. All SIP infants presented with a distended abdomen in adjunction to the timing of bowel perforation, known to occur early in the course of the disease [5]. Since lactate levels were measured prior to the day of diagnosis and clinical manifestation in the SIP group, the normal lactate range can be explained by presumed limited damage to the intestinal tissue at that time. Later on, post-perforation, lactate levels in the SIP diagnosed infants did increase. Furthermore, we suggest that the higher rate of thrombocytopenia observed in the SIP group might be correlated to the prenatal status of the dyad mother–infant (maternal pre-eclampsia [23] and intrauterine growth restriction (IUGR) [24]), being a residual effect seen in the early days of infant life. Hence, seven of the nine SIP infants with thrombocytopenia (78%) had either an SGA status, maternal pre-eclampsia or chorioamnionitis. Therefore, we mainly discussed the glucose and lactate findings in this study. As reported in the literature, infants in the SIP group were diagnosed on average within the first 5 days of life [6,8]. An additional validation of our data resides in the fact that all cases of SIP were surgically managed whereas expectant management was the choice in 75% of NEC cases, consistent with the literature [6,25,26]. The mortality rate in the SIP group was not statistically different than the one observed in the NEC group, and was within the range (19–39%) reported in the literature [3,14,27]. In a recent review by Swanson et al., the importance of improved definition and diagnosis of intestinal perforations has been discussed and a three out of five rule for bedside diagnosis of SIP has been proposed [28]. It included the following conditions: hyperglycemia within the 24 h prior to diagnosis, distended belly and up to 10 days of life at diagnosis [28]. As indicated by Pumberger et al., detected early, SIP can be treated by simple procedures (sutures, or resection and primary anastomosis) with a low rate of morbidity and mortality [6].

The study has a few limitations. First, it is limited by its retrospective nature and therefore prone to sampling bias. Secondly, the study is limited by a small sample size of the SIP group collected over 10 years at our institution. In addition, the data extraction procedure has shown that there were some discrepancies between indicated diagnosis (intestinal perforation) and clinical reports (medical NEC). Finally, despite a limited available data set, a low matching ratio, and our belief that surgical NEC infants (with perforation) would be a more suitable control group in the comparison with SIP infants, this pilot study provides valuable insights to warrant further studies.

## 5. Conclusions

Our results suggest that as opposed to NEC where lactate levels increased along with glucose levels, in SIP infants, prior to intestinal perforation as the onset of disease, hyperglycemia was not coupled to hyperlactatemia. We propose that a sudden hyperglycemia within the first week of life along with lactate levels within the normal range in VLBW and ELBW infants presenting with a distend belly can reveal a first clue on potential SIP development, similarly to what has been advocated by Swanson et al. [28].

## Figures and Tables

**Figure 1 children-10-01028-f001:**
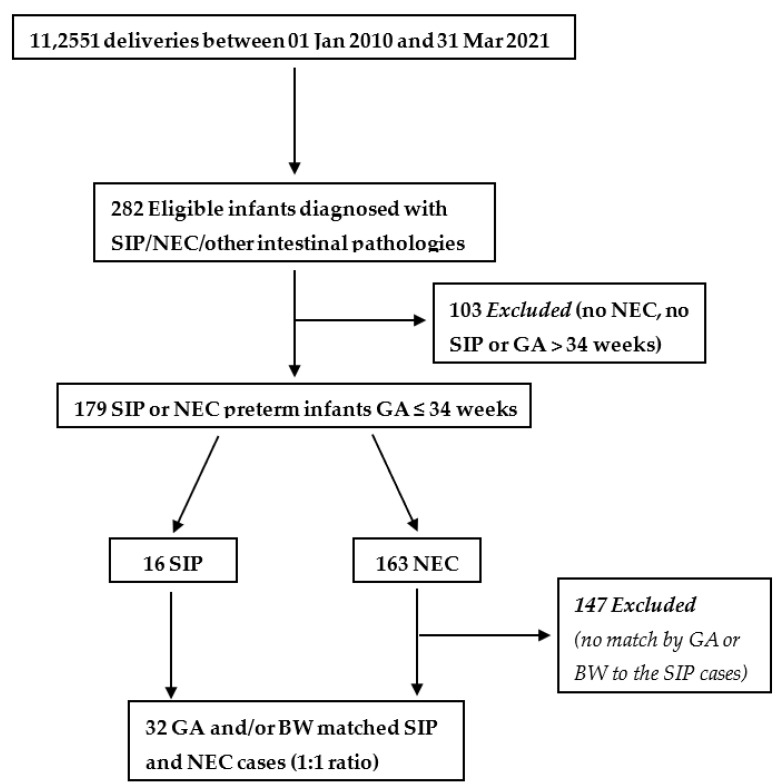
Flow diagram for the selection of infants. SIP, spontaneous intestinal perforation; NEC, necrotizing enterocolitis; GA, gestational age; BW, birth weight.

**Figure 2 children-10-01028-f002:**
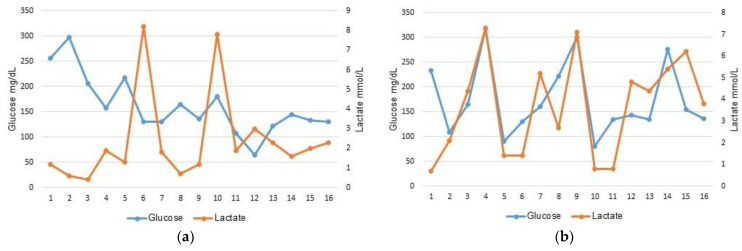
Correlation between glucose and lactate blood levels per group: (**a**) spontaneous intestinal perforation group; (**b**) necrotizing enterocolitis group.

**Table 1 children-10-01028-t001:** Maternal and neonatal characteristics.

Characteristics	SIP (*n* = 16)	NEC (*n* = 16)	*p*-Value
Maternal age, mean (SD), y	32.9 (5.3)	31.8 (6.8)	0.605
Pre-eclampsia	3 (18.8)	4 (25)	>0.99
Placental abruption	5 (31.3)	6 (37.5)	0.710
Chorioamnionitis	5 (31.3)	1 (6.3)	0.172
Antenatal steroids	14 (87.5)	14 (87.5)	>0.99
Magnesium sulfate	12 (75)	12 (75)	>0.99
Maternal antibiotics	13 (81.3)	13 (81.3)	0.685
Cesarean section	9 (56.3)	11 (68.8)	0.465
Gestational age, median (IQR), w	26.5 (25.6–28)	27 (25.2–28)	0.606
Birth weight, mean (SD), (range), g	978.1 (336.3) (540–1740)	950.9 (325) (525–1800)	0.818
Gender male	9 (56.3)	7 (43.8)	0.480
1 min Apgar score, mean (SD), (range)	4.9 (2.3) (1–9)	4.3 (2.4) (1–8)	0.503
5 min Apgar score, median (IQR)	7 (6.2–8)	7 (7–9.5)	0.558
AGA	13 (81.3)	11 (68.8)	0.414
LGA	0	0	
SGA	3 (18.8)	5 (31.3)	
Age at diagnosis, median (IQR), d	5 (3.2–7.8)	19 (11–29.5)	**<0.001**
NSAID (Ibuprofen)	5 (31.3)	8 (50)	0.280
Inotropic agent (Dopamine)	1 (6.3)	0	>0.99
Fresh Frozen Plasma (unit)	2 (12.5)	3 (18.8)	>0.99
Exclusive human milk feeding	11 (68.8)	13 (81.3)	0.685
Treatments			**<0.001**
Conservative	0	12 (75)	
Drain tube	4 (25)	0	
Primary closure	3 (18.8)	0	
Stoma	9 (56.2)	3 (18.8)	
Explorative laparotomy	0	1 (6.2)	
Death	5 (31.3)	4 (25)	>0.99
Hyperglycemia (Glucose > 120 mg/dL)	14 (87.5)	13 (81.3)	>0.99
Glucose, median (IQR), mg/dl	140 (131–199.8)	149 (132–230.2)	0.584
Hyperlactatemia (Lactate > 4 mmol/L)	2 (12.5)	8 (50)	**0.022**
Lactate, median (IQR), mmol/L	1.8 (1.2–2.3)	4.1 (1.4–5.6)	0.109
Platelets, median (IQR), K/µL	130.5 (97.2–184.8)	347.5 (186.8–510.2)	**<0.001**
Thrombocytopenia (<150 K/µL)	9 (56.3)	2 (12.5)	**0.009**
White blood cells, median (IQR), 10^3^ µL	9.7 (3–21.8)	14.3 (5.6–19.8)	0.366
^1^ NRBC, median (IQR), counts	1076 (243–2302)	404 (170–1409)	0.206

Data are expressed as *n* (%) or as indicated. Abbreviations: SIP: Spontaneous intestinal perforation; NEC: Necrotizing enterocolitis; AGA: appropriate for gestational age; LGA: large for gestational age; SGA: small for gestational age; NRBC: nucleated red blood cells. In bold, significant *p* values. ^1^ One missing in each group.

**Table 2 children-10-01028-t002:** Logistic regression analysis of laboratory parameters associated with spontaneous intestinal perforation.

					95% CI	
	B Coefficient	SE	OR	*p*-Value	Lower	Upper
Lactate	−0.499	0.243	0.607	0.040	0.377	0.978
Platelets	−0.018	0.007	0.982	0.011	0.969	0.996
Glucose	0.002	0.009	1.002	0.781	0.985	1.020

Abbreviations: SE, standard error; OR, odds ratio; CI, confidence interval.

## Data Availability

The datasets generated and analyzed are not publicly available but are available from the corresponding author upon reasonable request.

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
