# Peer review of "Blood Glucose, Lactate and Platelet Count in Infants with Spontaneous Intestinal Perforation versus Necrotizing Enterocolitis—A Pilot Study"

_children, 2023, doi:10.3390/children10061028_

Round 1
Reviewer 1 Report
This is a case control study that compared biomarkers (Glucose, lactate and platelet counts) among infants with a diagnosis of spontaneous intestinal perforation versus necrotizing enterocolitis
1)The sample size of the study (32 infants) would majorly limit inferences from such an analyses.
2)The authors could have matched the 16 cases of SIP with the 163 infants with NEC in a 1:3 or 1:4 ratio to utilize available information. Excluding 147 cases of NEC in this analysis at the cost of 1;1 case control match has lead to serious selection bias in this study.
3)The interpretations from logistic regression analysis may be incorrect. “The odds of SIP is reduced as the lactate level increases” is one such example. Commonly for such analysis, the reference group has to be infants with feed intolerance or suspected SIP/NEC.
Since the authors have compared SIP with a more serious condition such as NEC, they have ended up interpreting it in this manner,which can be misleading.
4) As a clinician , I would be interested in knowing if biomarkers can aid in differentiating a) surgical NEC from medical NEC and b) SIP from less severe abdominal pathologies. That would help in picking up serious intestinal pathologies at the earlier stage to institute definitive treatment
5)There is no information on the lactate, glucose and platelet count at baseline in the study patients.
Reviewer 2 Report
In this study the authors evaluated the possibility of finding indicators to direct the diagnosis towards SIP or NEC. The aim is good.
Line 66: for data better two investigators and after averaging between them
Line 75: explain NICU
Line 84: why Appropriate, small and large for gestational age newborns (AGA, SGA, LGA) were defined according to the Israeli-specific intrauterine growth charts? Not International?
Line 101-103: write better, not easy to understand, flow diagram is better
Round 2
Reviewer 2 Report
Ok